# LGBTQ+ Affirming Care May Increase Awareness and Understanding of Undetectable = Untransmittable among Midlife and Older Gay and Bisexual Men in the US South

**DOI:** 10.3390/ijerph191710534

**Published:** 2022-08-24

**Authors:** Tara McKay, Ellesse-Roselee Akré, Jeffrey Henne, Nitya Kari, Adam Conway, Isabel Gothelf

**Affiliations:** 1Department of Medicine, Health and Society, Vanderbilt University, Nashville, TN 37235, USA; 2Geisel School of Medicine, Dartmouth College, Hanover, NH 03755, USA; 3The Henne Group, Inc., San Francisco, CA 94107, USA

**Keywords:** affirming care, HIV, LGBTQ health, U = U, treatment as prevention, US

## Abstract

One of the most significant innovations in HIV prevention is the use of HIV treatment to prevent HIV transmission. This information has been disseminated as the “Undetectable = Untransmittable” (U = U) message. Despite evidence of effectiveness, U = U awareness, belief, and understanding remains limited in some communities. In this study, we examine whether having an LGBTQ affirming healthcare provider increases U = U awareness, belief, and understanding among midlife and older gay and bisexual men in the US south, an understudied and underserved population and region where new HIV infections are increasing. We used data from the Vanderbilt University Social Networks Aging and Policy Study (VUSNAPS) on sexual minority men aged 50 to 76 from four southern US states collected in 2020–2021. We found that only one in four men reported prior awareness of U = U, but awareness was higher among men who have an LGBTQ affirming provider. Among HIV negative men, those with an affirming provider were more likely to believe and understand U = U, have more accurate risk perception, and have ever tested for HIV. Improving access to LGBTQ affirming healthcare may improve U = U awareness, belief, and understanding, which could help to curb HIV transmission in the US south.

## 1. Introduction

One of the most significant innovations in HIV prevention in the last two decades has been the use of HIV treatment to prevent HIV transmission. The resulting global public health campaign, “Undetectable equals Untransmittable”, or U = U, underscores the importance of achieving and maintaining viral suppression in people living with HIV to prevent HIV transmission. By taking antiretroviral therapy (ART) daily as prescribed, people living with HIV cannot sexually transmit the virus to others [1]. In the US, this campaign has been widely supported by the Centers for Disease Control and Prevention, the National Institute of Allergy and Infectious Diseases, and the American Medical Association [1,2,3].

Large-scale survey studies using data collected from community-based samples of people living with HIV and men who have sex with men generally find very high levels of awareness of the U = U concept in the US [4,5] and observe increases in awareness of U = U over the last decade, especially among men living with HIV [5,6]. Among HIV negative men surveyed from 2017 and 2018 in the US, 85% report being aware of U = U [7]. 

However, several studies of men who have sex with men in the US suggest that understanding and application of U = U are substantially more limited. Online surveys of men who have sex with men consistently find that just three to four out of every ten men who have sex with men correctly identify HIV treatment or viral suppression as providing protection against transmission [8,9,10].

Importantly, existing studies have not generally focused on U = U awareness and understanding among midlife and older sexual minority men. The median age of most U = U awareness studies that include sexual minority men in the US is consistently younger than 40 and, for some, younger than 35 or even 30 [4,7,9,10]. One study that explicitly assessed age-cohort differences in HIV prevention knowledge, risk perception, and behaviors among gay and bisexual men in the US found that men in younger age-cohorts had greater functional knowledge of HIV prevention strategies, including condom use, pre-exposure prophylaxis (PrEP), post-exposure prophylaxis (PEP), and Treatment as Prevention/U = U [8]. 

Additionally, among studies with adequate sample sizes to test geographic variation within the US, HIV negative men living in southern states in the US were significantly less likely to have heard of U = U [7], and men living with HIV in southern states were less likely to rate the U = U concept as accurate compared to men living with HIV in the northeast and western states [11]. These gaps are important because southern states comprise more than 50% of new HIV infections, most of which are among men who have sex with men [12]. Additionally, fewer people in the south are aware that they are living with HIV compared to other US regions, delaying access to treatment, and there has been a lower uptake of other medical prevention technologies, such as PrEP [12,13]. 

In this study, we address gaps in the U = U landscape and expand on existing work by examining healthcare-related determinants of U = U awareness, understanding, and impact among midlife and older sexual minority men in the US south. Although many sexual minority men hear about the U = U message from sources other than a healthcare provider, healthcare providers remain an important source of information about U = U [4] and points of regular contact for HIV-positive men on treatment and HIV-negative men seeking sexual healthcare services or who are taking PrEP. However, a substantial share of sexual minority men do not talk to healthcare providers about sexual behavior and, thus, may not be receiving adequate sexual health care [14,15]. A recent national study found that, among men who have sex with men in the US, 30% of those with a primary care provider reported that they had not disclosed their sexual orientation to their primary healthcare provider [7]. 

Patients may not disclose their sexual behavior or identity for several reasons, including because providers do not ask, past negative experiences, fear of homophobia and stigmatization, internalized stigma, and belief that health is not related to their identity [16,17,18]. In a study of the Veterans Health Care Administration (VHA), more than one-third of gay, lesbian, and bisexual veterans (36.9%) reported that VHA staff “definitely does not know” about their sexual orientation and a quarter (25.1%) reported avoiding seeking services because of concerns about confidentiality, stigma, or acceptance of their sexual orientation [19]. 

There are also challenges to providing culturally and clinically appropriate care to LGBTQ people on the provider side. Today, the American Medical Association openly advocates for inclusion and nondiscrimination of LGBTQ+ patients and providers [20]. Although acceptance of LGBTQ people varies by physician specialty [21] and by reported versus implicitly held beliefs [22], studies generally find that many providers’ attitudes towards LGBTQ people are positive and have improved over time [23,24]. Nonetheless, many physicians still have difficulty providing LGBTQ affirming care—care that is respectful and meets the specific health needs of LGBTQ people--because they were not trained to do so. Half of US medical schools dedicate fewer than 5 hours to LGBTQ topics and decisions to include LGBTQ-related curricula at all are largely made by individual institutions [25,26]. As a result, many practitioners find it difficult to use unfamiliar sexual and gender terms, decide on which ward to nurse a transgender patient, discuss interpersonal violence and abuse with same-sex partners, and identify LGBTQ health care resources, despite otherwise holding positive attitudes toward LGBTQ people [16,27]. 

A lack of fluency in LGBTQ health, identities, and behaviors among providers can lead patients to delay or forgo care, even care that is not related to their LGBTQ identity or sexual health, and to not disclose their LGBTQ identities to providers [16,17,28]. Older sexual minority adults are particularly likely to report personal experiences or expectations of discrimination in healthcare settings, leading to delays in accessing care or forgone care [18]. For providers, patient nondisclosure or lack of comfort discussing sexual health issues can lead to the provision of inappropriate care, inattention to specific health care needs, missed diagnostic screenings, and less focus overall on creating LGBTQ inclusive healthcare environments in entire practices [29,30]. Specifically, for sexual health, sexual minority men who do not disclose their sexual orientation to their primary care provider are less likely to have been tested for HIV in the previous two years, less likely to have been tested for gonorrhea and syphilis, and less likely to have been vaccinated against hepatitis A and B [15,31,32]. The lack of affirming care options for sexual minorities can also lead to healthcare fragmentation. Gay and bisexual men often seek care from providers outside of primary care contexts, especially for their sexual health needs, because of gaps in provider knowledge, greater comfort with community providers, financial cost, or expectations of discrimination [33].

The barriers to accessing and providing LGBTQ affirming care may be particularly acute in southern US states. Southern states are more likely than northeastern and western states to have laws that explicitly exclude or do not provide adequate care for sexual and gender minorities in healthcare [34]. Southern states also have fewer “LGBTQ Healthcare Equality Leaders” compared to northeast and western states, according to the Human Rights Campaign 2020 Healthcare Equality Index [35]. LGBTQ affirming healthcare providers are more likely to have explicit employee and patient nondiscrimination policies as well as staff training in LGBTQ patient-centered care [35]. Lack of nondiscrimination policies perpetuates discriminatory behaviors such as verbal abuse and refusal to provide care, which deter patients and limit them from obtaining essential care [16]. 

Based on this prior work, we expect that gay and bisexual men with LGBTQ affirming providers may have different outcomes in relation to key sexual health and HIV prevention outcomes. We tested this hypothesis using original data on U = U awareness, understanding, and belief among midlife and older gay and bisexual men in four southern states in the US. 

## 2. Data and Methods

To examine the relationship among access to LGBTQ affirming care and knowledge and support of U = U among gay and bisexual men in the US south, we used data from the Vanderbilt University Social Networks, Aging, and Policy Study (VUSNAPS). VUSNAPS is a panel study of 1256 midlife and older LGBTQ adults aged 50 to 76 residing in Alabama, Georgia, North Carolina, and Tennessee. The VUSNAPS panel generally reflects the demographic characteristics of the LGBTQ population in the 50 to 76 age range for sample states and the US south as measured by the US Census Household Pulse Survey (HPS), Phase 3.2, weeks 34–39 (see Appendix A, Table A1). Compared to weighted HPS estimates of demographic characteristics of LGBTQ people in the US south census region, VUSNAPS participants are more educated, less likely to identify as bisexual, and less likely to identify as Latino/Hispanic. This study was approved by the Vanderbilt University Institutional Review Board. 

VUSNAPS participants were recruited using purposeful online and venue-based sampling, linked referral, and community outreach to organizations serving LGBTQ, men who have sex with men, and seniors in each state. Wave 1 was fielded from 1 April 2020 to 30 September 2021. In this study, we restricted analyses to VUSNAPS respondents who identify as gay or bisexual men assigned male at birth with complete information on all independent and dependent variables (N = 633).

### 2.1. U = U Measures

The VUSNAPS survey instrument includes several items to gauge awareness, belief, application, and impact of the U = U message. 

We measured awareness using the item, “Have you heard about U = U?” (1 = yes, 0 = no/don’t know). We also asked respondents who indicated having heard of U = U to identify where they heard of the concept. We also assessed general awareness with treatment as prevention using the item “I believe that HIV treatment makes people less likely to transmit the HIV virus”, rated on a 5-point Likert scale from 1 “disagree strongly” to 5 “agree very strongly”.

We measured belief in U = U using a detailed item that explains the U = U concept: “U = U means ‘Undetectable equals Untransmittable’, that is, HIV-positive people who take medication and bring their HIV viral load down to the point at which it is undetectable using standard medical tests cannot sexually transmit HIV to an HIV-negative partner. Please use the scale below to rate how much you believe the U = U concept”. Respondents were then presented with a Likert scale from 1 “Very unbelievable” to 5 “Very believable”. This item was recoded to 1 = ”Somewhat believable” or “Very believable” versus all others = 0.

To measure understanding of U = U, participants read the following vignette: “Please imagine a situation in which an HIV-positive man, who is taking medications and reduced his viral load to a point where it is undetectable, has unprotected anal sex with an HIV-negative man. The HIV-positive man is the top and he ejaculates inside the HIV-negative man. How likely do you think it is that the HIV-negative man will get the HIV virus from this encounter?” Response categories included a 7-point Likert scale from 1 “No chance or almost no chance”, 2 “Very unlikely”, 3 “Somewhat unlikely”, 4 “Not sure”, 5 “Somewhat likely”, 6 “Very likely”, 7 “Certain or almost certain”. For logistic analyses, we collapse responses “No chance” and “Very unlikely = 1 (correct understanding) versus all others = 0.

Finally, we measured the impact of U = U on risk perception using the item “I would feel safe having sex with someone who is HIV-positive as long as he is receiving treatment and has reduced his viral load to a point where it is undetectable”. Respondents rated their level of agreement on a 5-point Likert scale from 1 “Strongly disagree” to 5 “Strongly agree”. For logistic analyses, we collapsed “Agree” and “Strongly agree” = 1 versus all others = 0.

### 2.2. Affirming Care Measures

The VUSNAPS instrument includes several measures to assess healthcare access and utilization, including whether the respondent has a usual source of care, whether the respondent has a primary or secondary health care provider whom they perceive as LGBTQ affirming, and reasons for not having an LGBTQ affirming provider if no affirming provider is identified.

To assess access to an LGBTQ affirming provider, respondents were asked, “Do you have an LGBT affirming health care provider?” with response options: “Yes, they are my primary health care provider”; “Yes, I see them in addition to another health care provider”; “No, I don’t need or want an LGBT affirming health care provider”; “No, I cannot find an LGBT affirming health care provider in my area”; “I don’t know”; and “No answer”. Respondents who reported “Yes” were coded as having access to an LGBTQ affirming health care provider. No answer was coded as missing. All others were coded as no. 

### 2.3. Covariates

We included demographic characteristics including age, education (high school or less, some college, college degree, and graduate or professional degree), partnered, race/ethnicity (white, black, other\multiracial), HIV status, and, for HIV negative men, whether they have ever had an HIV test. 

### 2.4. Analytic Strategy

We conduct descriptive analyses and logistic regression analyses stratified by HIV status for each U = U outcome identified above, controlling for state of residence, age, education, race/ethnicity, whether the respondent has a partner or spouse, whether the respondent reported hearing of U = U prior to the survey, and whether they have ever tested for HIV (HIV negative men only). 

## 3. Results

Table 1 presents the distribution of the sample across demographic and geographic characteristics for gay and bisexual men in the VUSNAPS sample stratified by HIV status. Men living with HIV were significantly more likely to have an LGBTQ affirming care provider, to be Black or other\multiracial compared to White, and to have less education than men who were HIV negative. 

### 3.1. LGBTQ Affirming Care

About two of every three men in the study (64.8%) identified a primary or secondary healthcare provider as LGBTQ affirming. Just over half of HIV negative men (59.0%) identified their healthcare provider as LGBTQ affirming compared with almost all (87.3%) of men living with HIV. After adjusting for state of residence and demographic characteristics (age, education, race/ethnicity), men living with HIV were more than seven times more likely to identify their healthcare provider as LGBTQ affirming compared with HIV negative men (OR = 7.10; 95% CI = 3.94–12.80; see Table 2). Having an LGBTQ affirming care provider increased the odds that HIV negative men reported ever testing for HIV by more than two times (OR = 2.26; 95% CI = 1.38–3.72).

### 3.2. Awareness of U = U

The most well-studied dimension of the U = U concept is sexual minority men’s awareness of U = U. Only about one in four (25.4%) gay and bisexual men in the four sample states had previously heard of the U = U concept specifically; a majority (70.5%) had not heard of the U = U concept or were uncertain (4.4%). Men living with HIV (56.3%) were significantly more likely to have heard of U = U specifically compared with HIV negative men (17.5%; χ2 = 85.83; *p* < 0.001). Both HIV negative men and men living with HIV who had an LGBTQ affirming provider were significantly more likely to have heard about U = U specifically compared to men of the same HIV status without an affirming provider (*p* < 0.001; see Table 3). While most had not heard of U = U specifically, a larger share of gay and bisexual men (76.0%) was generally aware of the idea of treatment as prevention—that HIV treatment makes people less likely to transmit the virus. Again, men were more likely to have general awareness of U = U when they had an LGBTQ+ affirming provider (χ2 = 11.24; *p* ≤ 0.001).

In logistic regression analyses adjusting for other respondent characteristics and geographic variation (see Table 4), we found that HIV negative men with an affirming care provider were more than three times more likely to have heard of U = U (OR = 3.13; 95% CI = 1.75–5.61). Even among men living with HIV, who had high overall awareness of U = U, those with an affirming care provider were almost five times more likely to report having heard of U = U (OR = 4.87; 95% CI = 1.31–18.05).

Among those who had heard of U = U specifically (N = 160), the most common sources of initial information were 1) the internet (26.3%), 2) a television or print advertisement or story (23.1%), or 3) a health care provider (19.4%). Respondents also indicated that they had heard about U = U from community outreach and meetings, family or friends, and social networking or dating apps. We observed significant differences in how participants heard about U = U by whether the respondent also reported having an LGBTQ affirming healthcare provider (χ2 = 18.11; *p* < 0.01). Among those with an LGBTQ affirming provider, 21.2% heard about U = U from their healthcare provider compared with just 8.7% among those not reporting an LGBTQ affirming healthcare provider. Among HIV negative men, those who reported hearing about U = U from a healthcare provider all indicated that their provider was LGBTQ affirming. Sample size limitations prohibit us from further disaggregation or adjusted analyses of information sources. 

We found similar results for general awareness of treatment as prevention among HIV negative gay and bisexual men. In logistic regression analyses adjusting for other respondent characteristics and geographic variation (see Table 5), we found that HIV negative men with an affirming care provider were almost two times more likely to be aware of the idea of treatment as prevention (OR = 1.78; 95% CI = 1.16–2.72).

### 3.3. Believability of U = U

Following a short description of the U = U concept, a majority of sexual minority men rated U = U as “very believable” (35.1%) or “somewhat believable” (26.5%). About a quarter (23.4%) were unsure and a nontrivial minority rated U = U as “very unbelievable” (5.8%) or somewhat unbelievable (9.3%). In bivariate analyses, individuals with an LGBTQ affirming care provider were significantly more likely than individuals without an affirming care provider to rate U = U as “somewhat” or “very believable” (67.0% versus 51.6%; χ2 = 31.35; *p* < 0.001). This difference remains when bivariate analyses are restricted to just HIV negative men (63.3% vs. 49.0%; χ2 = 16.37; *p* < 0.001).

After controlling for other demographic and geographic factors in an adjusted logistic regression model (see Table 6), we found that HIV negative men with an LGBTQ affirming provider are one-and-a-half times more likely to rate the U = U concept as “somewhat” or “very believable” (OR = 1.53; 95% CI = 1.02–2.30).

### 3.4. Understanding of U = U

Participants were asked to apply the U = U concept to assess the likelihood that an HIV negative man would contract HIV in a hypothetical, condomless sexual encounter with a man living with HIV who was on treatment and had an undetectable viral load. Individuals with an LGBTQ affirming provider were more likely to correctly identify that the HIV negative man had “no chance or almost no chance” of contracting the virus during the described sexual encounter (26.3% vs. 9.4%; χ2 = 39.39; *p* < 0.001). Although men living with HIV were more likely to correctly understand the U = U concept in this context, the gap in understanding by whether individuals had an affirming care provider was present for both HIV negative men and men living with HIV.

In adjusted logistic regression analyses (see Table 7), we found that HIV negative men with an LGBTQ affirming care provider were about one-and-a-half times more likely to understand and correctly apply the U = U concept to a hypothetical scenario significant at the *p* < 0.1 level (OR = 1.45; 95% CI = 0.96–2.20). 

### 3.5. Impact of U = U on Risk Perception

To assess the impact of U = U on perceived risk, we asked participants to rate their level of agreement with the statement “I would feel safe having sex with someone who is HIV-positive as long as they are receiving treatment and have reduced their viral load to a point where it is undetectable”. A majority (57.0%) of sexual minority men agreed or strongly agreed with this statement. HIV positive men (87.9%) were significantly more likely to agree or strongly agree compared with HIV negative men (49.1%; χ2 = 64.42; *p* < 0.001). Among HIV negative men, those with an LGBTQ affirming care provider (56.1%) were significantly more likely to view having sex with someone who is HIV positive and undetectable as “safe” compared with HIV negative men who did not report an LGBTQ affirming provider (37.0%; χ2 = 17.66; *p* < 0.001). After controlling for other respondent demographic characteristics and geographic location (see Table 8), we found that having an LGBTQ affirming care provider increased the odds of feeling safe having sex with someone who is HIV positive and undetectable by almost two-and-a-half times (OR = 2.02; 95% CI = 1.33–3.05).

## 4. Discussion

In this paper, we examined the relationship between having an LGBTQ affirming provider and several U = U related outcomes, including awareness, belief, understanding, and impact on risk perception. About two-thirds of sexual minority men in the study reported having an LGBTQ affirming healthcare provider as their primary or secondary provider. Unsurprisingly, men living with HIV were several times more likely to report having an LGBTQ affirming healthcare provider compared with HIV negative men. 

Strikingly, the midlife and older gay and bisexual men in the US south surveyed by VUSNAPS were largely unaware of “U = U” and the “undetectable = untransmittable” language, even though a majority were generally familiar with the idea of treatment as prevention. This was especially true of HIV negative men in this study, only 17.5% of whom reported being aware of U = U specifically prior to the study. Awareness of U = U in this study was substantially lower than in other international and US surveys of people living with HIV and men who have sex with men that find high (70–90%) awareness of U = U [4,5,7]. Higher awareness in other studies may be due to sampling strategies that primarily engage individuals attached to organizations for people living with HIV, use of measures that assess general understanding of treatment as prevention rather than U = U specific awareness, or lack of disaggregation of HIV negative from men living with HIV in some samples. In other work, HIV negative and unknown status men who have sex with men are significantly less likely to be aware of U = U [5]. Additionally, VUSNAPS is a study of midlife and older LGBTQ adults aged 50 to 76 in a region that is disproportionately growing in HIV cases relative to the rest of the US [12], has fewer HIV and LGBTQ affirming providers [35], has more rural and suburban LGBTQ adults [36,37], and has poorer access to healthcare overall [38]. Unlike other convenience sample studies, VUSNAPS purposefully recruited from a range of online and community venues that included but were not limited to HIV and LGBTQ community organizations, and, thus, the sample may reflect a population that is less well-connected to HIV care and information. Unlike other samples of substantially younger men who have sex with men, VUSNAPS also focuses exclusively on midlife and older sexual and gender minority populations in the US south.

On all U = U outcomes—awareness, belief, understanding, and impact on risk perception—we observe that HIV negative men with an affirming care provider have a greater likelihood of a positive outcome (see Figure 1). This result is replicated when examining general awareness of treatment as prevention, the idea behind the U = U message. HIV negative men with an LGBTQ affirming care provider are also more than two times more likely to have ever received an HIV test compared to HIV negative men without an affirming care provider. Importantly, we also observe significant improvement in awareness of the U = U concept among men living with HIV who have an LGBTQ affirming care provider compared with men living with HIV who do not have an affirming care provider.

There may be several mechanisms that produce these improved U = U outcomes. We find that those with an LGBTQ affirming care provider were more likely to have heard about U = U from a healthcare provider. This finding is consistent with broader findings that sexual minority patients are more likely to communicate about their specific health needs and behaviors in affirming care contexts [39]. LGBTQ affirming providers may also be more comfortable having conversations about HIV and sexual health with sexual minority men.

These findings have important implications for clinical guidance and medical education. Most physicians are comfortable treating gay patients, especially more recent medical school graduates [23,24,28,40]. However, additional clinical training and medical education courses on how to provide LGBTQ affirming care would likely decrease gaps in U = U awareness and understanding and may increase HIV testing among midlife and older HIV negative men in the US south. Other institution and practice level changes, such as the adoption of explicit nondiscrimination policies for patients and employees and the use of inclusive language and images in health care settings, may improve LGBTQ patient outcomes. In this study, healthcare providers were among the top three sources of information about U = U, and men reporting an LGBTQ affirming provider were significantly more likely to have heard about U = U from their healthcare provider. 

Improving access to and provision of LGBTQ affirming care among sexual minority men may also reduce HIV stigma within the LGBTQ community. We find that having an LGBTQ affirming care provider increased the odds of feeling safe having sex with someone who is living with HIV and has an undetectable viral load by almost two-and-a-half times (OR = 2.02; 95% CI = 1.33–3.05). Decreasing HIV stigma is important for the well-being of men living with HIV and increases testing among HIV negative men [41].

This study has some limitations. First, assessment of whether respondents had access to an LGBTQ affirming provider did not provide a definition or example. Future work should unpack the behaviors or cues that underpin patient perceptions of affirming or nonaffirming care. Second, while we see strong signals of the effects of having an LGBTQ affirming provider for HIV negative men, we lack the power to assess differences among men living with HIV, the vast majority of whom report an LGBTQ affirming provider as their primary or secondary healthcare provider. We are also unable to disaggregate experiences across race/ethnicity and sexual identity among HIV negative men. New HIV infections in southern states are growing fastest among Black men who have sex with men. Our findings suggest but cannot confirm that greater access to LGBTQ affirming care would be particularly beneficial for increasing U = U awareness and HIV testing among Black sexual minority men in the south. 

## 5. Conclusions

The southern region has the greatest burden of HIV-related deaths in the US. Compared with those unaware of U = U, people living with HIV who have U = U-related discussions with a health care provider have better odds of adherence to HIV treatments and viral suppression [4]. This paper demonstrates that expanding LGBTQ affirming care may help reduce HIV-related mortality in the south by improving U = U awareness and uptake of HIV testing among midlife and older HIV negative men.

## Figures and Tables

**Figure 1 ijerph-19-10534-f001:**
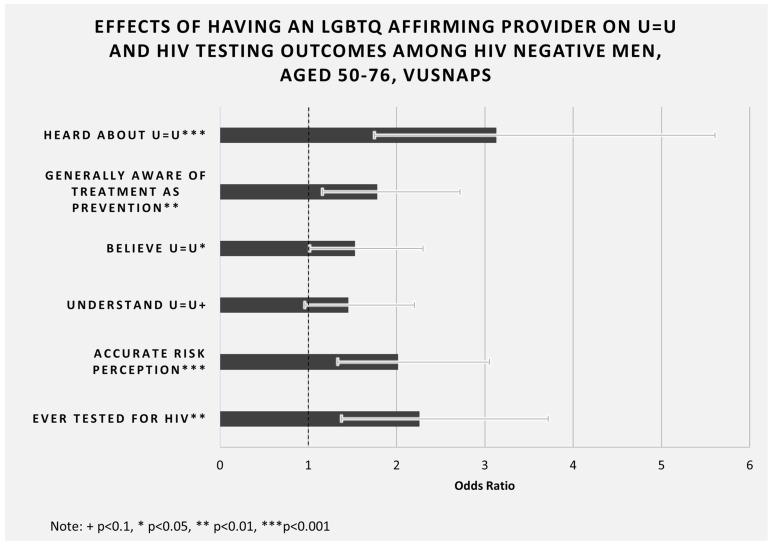
Effects of having an LGBTQ affirming provider on U = U and HIV testing outcomes among HIV negative men, aged 50--76, VUSNAPS. Note: Adjusted odds ratios are presented for HIV negative men only from the analyses presented in Table 2, Table 4, Table 5, Table 6 and Table 7. Analyses control for state of residence, age, education, race/ethnicity, partner status, ever tested for HIV (except for models where this is the outcome), and ever heard of U = U (except for models where this is the outcome). All odds ratios are significant at the *p* < 0.05 level or higher, except “Understand U = U”, which is significant at the *p* < 0.1 level.

**Table 1 ijerph-19-10534-t001:** Characteristics of older gay and bisexual men in the US south by HIV status, VUSNAPS.

	Overall (N = 633)		HIV Negative Men (N = 502)		Men Living with HIV (N = 131)		*p*
	N	%	N	%	N	%	
**Has LGBTQ Affirming Provider**	413	65.2	299	59.6	114	87.0	***
**Ever Tested for HIV**	544	85.9	413	82.3	138	100.0	***
**State of Residence**							
Alabama	106	16.7	83	16.5	23	17.6	
Georgia	161	25.4	119	23.7	42	32.1	
North Carolina	176	27.8	149	29.7	27	20.6	
Tennessee	190	30.0	151	30.1	39	29.8	
**Age (M/sd)**	59.4	6.30	59.7	6.4	58.5	6.0	
**Education**							**
High School or less	29	4.6	18	3.6	11	8.4	
Some College	145	22.9	106	21.1	39	29.8	
College Degree	218	34.4	173	34.5	45	34.4	
Graduate/Professional Degree	241	38.1	205	40.8	36	27.5	
**Race/Ethnicity**							***
White	559	88.3	460	91.6	99	75.6	
Black	38	6.0	20	4.0	18	13.7	
Other\Multiracial	36	5.7	22	4.4	14	10.7	
**Partnered**	382	60.3	329	65.5	53	40.5	***

Note: Bivariate comparisons are conducted using chi-square tests for categorical variables and independent samples *t*-test for continuous variables. ** *p* < 0.01, *** *p*< 0.001.

**Table 2 ijerph-19-10534-t002:** Odds of reporting an LGBTQ affirming provider and ever testing for HIV among a sample of midlife and older gay and bisexual men in the US south, VUSNAPS.

	Has LGBTQ Affirming Care Provider (N = 633)		Ever Tested for HIV (HIV Negative Men Only; N = 502)	
	OR	95% CI	OR	95% CI
Living with HIV	7.10 ***	(3.94–12.80)		
Has LGBTQ Affirming Provider	--		2.260 **	(1.38–3.72)
State of Residence				
Georgia	1.00	(1.00–1.00)	1.00	(1.00–1.00)
Alabama	0.82	(0.48–1.42)	0.89	(0.43–1.85)
North Carolina	1.58 +	(0.96–2.58)	1.28	(0.65–2.49)
Tennessee	1.16	(0.72–1.85)	1.38	(0.71–2.68)
Age	0.99	(0.96–1.01)	0.97	(0.93–1.01)
Education				
High School or less	1.00	(1.00–1.00)	1.00	(1.00–1.00)
Some College-	2.01	(0.82–4.93)	0.41	(0.11–1.56)
College Degree	2.53 *	(1.06–6.15)	1.21	(0.31–4.70)
Graduate/Professional Degree	4.62 ***	(1.90–11.21)	0.78	(0.20–3.02)
Race/Ethnicity				
White	1.00	(1.00–1.00)	1.00	(1.00–1.00)
Black	0.86	(0.39–1.89)	1.14	(0.31–4.18)
Other\Multiracial	0.59	(0.27–1.26)	0.43 +	(0.16–1.15)
Partnered	1.72 **	(1.19–2.48)	0.68	(0.40–1.16)
pseudo R-sq	0.10		0.07	

Note: Odds ratios are exponentiated logit coefficients. + *p* < 0.1, * *p* < 0.05, ** *p* < 0.01, *** *p* < 0.001.

**Table 3 ijerph-19-10534-t003:** Characteristics of midlife and older gay and bisexual men in the US south by HIV status, VUSNAPS.

	Full Sample (N = 633)						
	Overall		No Affirming Care		Affirming Care		
	N	%	N	%	N	%	*p*
Heard of U = U Prior to Survey	160	25.3	24	10.9	136	32.9	***
Generally Aware of Treatment as Prevention	475	76.0	148	67.3	327	79.4	***
U = U Believable	391	61.8	114	51.8	277	67.1	***
Correct Application of U = U	303	47.9	76	34.5	227	55.0	***
U = U Decreases Perception of Risk	351	56.3	88	40.0	263	63.8	***
Total	633	100.0	220	100.0	413	100.0	
	**HIV Negative Men (N = 502)**						
	**Overall**		**No Affirming Care**		**Affirming Care**		
	**N**	**%**	**N**	**%**	**N**	**%**	** *p* **
Heard of U = U Prior to Survey	87	17.3	19	9.4	68	22.7	***
Generally Aware of Treatment as Prevention	362	71.7	135	65.5	227	75.9	*
U = U Believable	290	57.8	100	49.3	190	63.5	***
Correct Application of U = U	207	41.2	66	32.5	141	47.2	***
U = U Decreases Perception of Risk	238	47.4	73	36.0	165	55.2	***
Total	502	100.0	203	100.0	299	100.0	
	**Men Living with HIV (N = 131)**						
	**Overall**		**No Affirming Care**		**Affirming Care**		
	**N**	**%**	**N**	**%**	**N**	**%**	** *p* **
Heard of U = U Prior to Survey	73	55.7	5	29.4	68	59.6	*
Generally Aware of Treatment as Prevention	102	77.3	14	82.4	88	76.5	
U = U Believable	101	77.1	14	82.4	87	76.3	
Correct Application of U = U	96	73.3	10	58.8	86	75.4	
U = U Decreases Perception of Risk	113	88.3	15	88.2	98	88.3	
Total	131	100.0	17	100.0	114	100.0	

Note: Bivariate comparisons are conducted using chi-square tests for categorical variables. * *p* < 0.05, *** *p* < 0.001.

**Table 4 ijerph-19-10534-t004:** Estimates of U = U awareness among a sample of midlife and older gay and bisexual men in the US south by HIV status, VUSNAPS.

	Heard about U = U			
	HIV Negative Men (N = 502)		Men Living with HIV (N = 131)	
	OR	95% CI	OR	95% CI
Has LGBTQ Affirming Provider	3.13 ***	(1.75–5.61)	4.87 *	(1.31–18.05)
Ever Tested for HIV	1.82	(0.84–3.93)	--	
State of Residence				
Georgia	1.00	(1.00–1.00)	1.00	(1.00–1.00)
Alabama	1.15	(0.55–2.41)	3.31 +	(0.88–12.49)
North Carolina	0.56 +	(0.29–1.11)	0.80	(0.27–2.38)
Tennessee	0.67	(0.35–1.29)	0.51	(0.19–1.35)
Age	0.98	(0.95–1.02)	1.01	(0.94–1.08)
Education				
High School or less	1.00	(1.00–1.00)	1.00	(1.00–1.00)
Some College	4.45	(0.54–36.66)	3.09	(0.68–14.09)
College Degree	2.69	(0.33–21.82)	0.87	(0.20–3.84)
Graduate/Professional Degree	3.71	(0.46–29.85)	1.55	(0.34–7.09)
Race/Ethnicity				
White	1.00	(1.00–1.00)	1.00	(1.00–1.00)
Black	1.94	(0.67–5.58)	1.01	(0.32–3.26)
Other\Multiracial	1.45	(0.45–4.65)	3.14	(0.72–13.59)
Partnered	0.67	(0.40–1.13)	3.48 **	(1.43–8.47)
pseudo R-sq	0.074		0.156	

Note: Odds ratios are exponentiated logit coefficients. + *p* < 0.1, * *p* < 0.05, ** *p* < 0.01, *** *p* < 0.001.

**Table 5 ijerph-19-10534-t005:** Estimates of general awareness of treatment as prevention among a sample of midlife and older gay and bisexual men in the US south by HIV status, VUSNAPS.

	General Awareness of Treatment as Prevention			
	HIV Negative Men (N = 502)		Men Living with HIV (N = 131)	
	OR	95% CI	OR	95% CI
Has LGBTQ Affirming Provider	1.78 **	(1.16–2.72)	0.14	(0.01–1.65)
Ever Tested for HIV	1.73 *	(1.04–2.88)	--	
State of Residence				
Georgia	1.00	(1.00–1.00)	1.00	(1.00–1.00)
Alabama	1.05	(0.53–2.08)	0.75	(0.09–6.13)
North Carolina	0.90	(0.50–1.62)	0.20 +	(0.03–1.20)
Tennessee	0.45 **	(0.26–0.79)	0.35	(0.06–1.94)
Age	0.98	(0.95–1.01)	1.01	(0.91–1.14)
Education				
High School or less	1.00	(1.00–1.00)	1.00	(1.00–1.00)
Some College	0.92	(0.29–2.91)	1.54	(0.20–11.86)
College Degree	0.92	(0.30–2.83)	2.51	(0.32–19.62)
Graduate/Professional Degree	1.08	(0.35–3.33)	2.32	(0.28–19.06)
Race/Ethnicity				
White	1.00	(1.00–1.00)	1.00	(1.00–1.00)
Black	0.65	(0.24–1.77)	0.10 **	(0.02–0.41)
Other\Multiracial	1.04	(0.98–2.84)	0.36	(0.06–2.26)
Partnered	0.65 +	(0.42–1.02)	0.49	(0.14–1.71)
pseudo R-sq	0.049		0.171	

Note: Odds ratios are exponentiated logit coefficients. + *p* < 0.1, * *p* < 0.05, ** *p* < 0.01.

**Table 6 ijerph-19-10534-t006:** Estimates of U = U belief among a sample of midlife and older gay and bisexual men in the US south by HIV status, VUSNAPS.

	Believe U = U			
	HIV Negative Men (N = 502)		Men Living with HIV (N = 131)	
	OR	95% CI	OR	95% CI
Has LGBTQ Affirming Provider	1.53 *	(1.02–2.30)	0.41	(0.09–1.86)
Heard of U = U Prior to Survey	8.06 ***	(3.75–17.31)	1.53	(0.57–4.10)
Ever Tested for HIV	1.39	(0.84–2.30)	--	
State of Residence				
Georgia	1.00	(1.00–1.00)	1.00	(1.00–1.00)
Alabama	0.68	(0.36–1.28)	0.24 +	(0.06–1.03)
North Carolina	0.61 +	(0.36–1.05)	0.28 +	(0.07–1.16)
Tennessee	0.52 *	(0.31–0.89)	0.24 *	(0.07–0.87)
Age	0.98	(0.95–1.01)	1.00	(0.92–1.08)
Education				
High School or less	1.00	(1.00–1.00)	1.00	(1.00–1.00)
Some College	1.71	(0.59–4.98)	0.27	(0.03–2.55)
College Degree	1.46	(0.52–4.09)	0.28	(0.03–2.63)
Graduate/Professional Degree	1.69	(0.60–4.77)	0.52	(0.05–5.39)
Race/Ethnicity				
White	1.00	(1.00–1.00)	1.00	(1.00–1.00)
Black	0.81	(0.29–2.24)	0.47	(0.14–1.66)
Other\Multiracial	1.70	(0.64–4.50)	0.41	(0.11–1.54)
Partnered	0.82	(0.54–1.25)	0.72	(0.28–1.83)
pseudo R-sq	0.11		0.10	

Note: Odds ratios are exponentiated logit coefficients. + *p* < 0.1, * *p* < 0.05, *** *p* < 0.001.

**Table 7 ijerph-19-10534-t007:** Estimates of understanding of U = U among a sample of midlife and older gay and bisexual men in the US south by HIV status, VUSNAPS.

	Understands U = U			
	HIV Negative Men (N = 502)		Men Living with HIV (N = 131)	
	OR	95% CI	OR	95% CI
Has LGBTQ Affirming Provider	1.45 +	(0.96–2.20)	1.50	(0.36–6.23)
Heard of U = U Prior to Survey	3.58 ***	(2.11–6.06)	9.41 ***	(3.04–29.13)
Ever Tested for HIV	1.31	(0.77–2.22)	--	
State of Residence				
Georgia	1.00	(1.00–1.00)	1.00	(1.00–1.00)
Alabama	0.69	(0.38–1.28)	0.24 +	(0.06–1.04)
North Carolina	1.01	(0.60–1.71)	0.79	(0.19–3.24)
Tennessee	0.43 **	(0.25–0.73)	0.46	(0.14–1.51)
Age	0.99	(0.96–1.02)	1.00	(0.92–1.09)
Education				
High School or less	1.00	(1.00–1.00)	1.00	(1.00–1.00)
Some College	1.89	(0.56–6.40)	1.09	(0.20–5.85)
College Degree	1.72	(0.52–5.64)	2.47	(0.48–12.89)
Graduate/Professional Degree	2.41	(0.73–7.91)	2.30	(0.40–13.17)
Race/Ethnicity				
White	1.00	(1.00–1.00)	1.00	(1.00–1.00)
Black	0.57	(0.20–1.59)	0.48	(0.14–1.67)
Other\Multiracial	0.25 *	(0.08–0.78)	6.41	(0.58–70.79)
Partnered	0.88	(0.58–1.34)	1.04	(0.36–3.00)
pseudo R-sq	0.09		0.23	

Note: Odds ratios are exponentiated logit coefficients. + *p* < 0.1, * *p* < 0.05, ** *p* < 0.01, *** *p* < 0.001.

**Table 8 ijerph-19-10534-t008:** Estimates of impact of U = U on risk perception among a sample of midlife and older gay and bisexual men in the US south by HIV status, VUSNAPS.

	U = U Decreases Perception of Risk			
	HIV Negative Men (N = 495)		Men Living with HIV (N = 128)	
	OR	95% CI	OR	95% CI
Has LGBTQ Affirming Provider	2.02 ***	(1.33–3.05)	0.25	(0.03–2.27)
Heard of U = U Prior to Survey	4.28 ***	(2.41–7.60)	3.93 +	(0.82–18.94)
Ever Tested for HIV	1.96 *	(1.15–3.35)	--	
State of Residence				
Georgia	1.00	(1.00–1.00)	1.00	(1.00–1.00)
Alabama	0.72	(0.39–1.34)	0.57	(0.06–5.85)
North Carolina	0.64	(0.38–1.09)	0.27	(0.04–2.04)
Tennessee	0.54 *	(0.32–0.92)	0.26	(0.04–1.53)
Age	0.98	(0.95–1.01)	1.02	(0.90–1.16)
Education				
High School or less	1.00	(1.00–1.00)	1.00	(1.00–1.00)
Some College	1.68	(0.55–5.18)	0.18	(0.02–1.95)
College Degree	1.58	(0.53–4.68)	1.02	(0.08–13.15)
Graduate/Professional Degree	1.43	(0.48–4.25)	1.31	(0.09–19.28)
Race/Ethnicity				
White	1.00	(1.00–1.00)	1.00	(1.00–1.00)
Black	1.30	(0.48–3.55)	0.18 *	(0.04–0.86)
Other\Multiracial	0.80	(0.31–2.03)	0.72	(0.06–8.07)
Partnered	0.72	(0.48–1.10)	1.65	(0.36–7.61)
pseudo R-sq	0.10		0.23	

Note: Odds ratios are exponentiated logit coefficients. + *p* < 0.1, * *p* < 0.05, *** *p* < 0.001.

## Data Availability

The data presented in this study are available on request from the corresponding author.

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
