# Peer review of "LGBTQ+ Affirming Care May Increase Awareness and Understanding of Undetectable = Untransmittable among Midlife and Older Gay and Bisexual Men in the US South"

_ijerph, 2022, doi:10.3390/ijerph191710534_

Round 1

Reviewer 1 Report

Thank you for the opportunity to review this submission. This article, entitled "LGBTQ+ Affirming Care May Increase Awareness and Understanding of Undetectable=Untransmittable among Midlife and Older Gay and Bisexual Men in the US South," assesses the relationship between having an affirming PCP and U=U awareness, belief, and understanding among older queer men in four southern states. Overall, the article is very well written, fills a gap in the literature, and has direct clinical and programmatic implications. 

The following minor revisions would further enhance the quality and clarity of the article ahead of publication and dissemination: 

Overall: 

1. Recommend using consistent language throughout, specifically around the age range of the population and terminology describing HIV status. The title and beginning of the article refer to "midlife and older" men, but then most of the tables and results only speak to "older" men. I would also recommend consistency around "living with HIV" - see line 33 for example of "people with HIV" or line 63 for example of "they have HIV" language. 

Introduction: 

2. Line 57 introduces "studies" demonstrating that HIV-negative men in the South are significantly less likely to have heard of U=U, however only one study is cited. Additional references are needed to support that multiple studies have demonstrated this finding, otherwise suggest revising the language here. 

Methods: 

3. Line 132 - might suggest another term besides "use" of LGBTQ affirming care. The current language might imply affirming care is readily available and uptake is the issue. Possibly "receipt of" or "provision of" LGBTQ affirming care instead. 

4. Typo in Line 149 with duplicate "to."

5. Additional information would be beneficial under the Affirming Care Measures section. How was an affirming PCP quantified or asked in the assessment? Were possible reasons for not having an affirming provider listed for selection, or were the responses unprompted? More clarity and specificity, similar to that included in the U=U measures section, would strengthen the article since these measures are a primary focus of analysis. 

Results: 

6. For all tables, clarity would be improved by left aligning the variable names and indenting subcategories as appropriate. For example, indenting the four state names under "State of Residence" or the educational attainment level under "Education." Might also recommend, at the discretion of the authors, that the overall and stratified N's be added to the headers (similar to Table 1) instead of the last row for Tables 2-7. 

7. Reduce the number of decimal places reported in the Tables, and be consistent between the Tables and in-text statistics in terms of reported decimal places.

8. In Table 1, prior HIV testing behaviors are reported for HIV-negative men only (n=441), but then the overall frequency is different (n=556). It's unclear where the additional 115 responses in the Overall frequency are coming from. 

9. Inclusion of chi-square statistics in Table 3 would be beneficial, as only one chi-square test is reported in-text. Additionally, the percentage in the text (17.5%) is slightly different than the percentage in the table (17.7%). 

10. If space/word count allows, consider speaking to other statistically significant relationships, as noted in the Tables, in the text. 

Discussion: 

11. Line 298 refers to findings from the broader literature, but no citations are included.

12. It would be beneficial to speak to the representativeness of the sample, either in the Results or the Limitations. For example, the sample included in the analysis is largely White, educated, and in the lower end of the eligible age range. Are data about recruitment strategies available to report? For example, were most men recruited from online platforms or in-person venues? Is there the potential for bias based on these data? 

13. The last paragraph (Lines 350-355), especially the last sentence, seem somewhat removed and generalized from the direct findings of the paper. Given the significant findings found from a region disproportionately burdened by HIV and with numerous disparities in terms of equitable and quality care, the currently worded closing paragraph seems like a missed opportunity to really highlight the implications, clinical or policy recommendations, and/or next steps from your research. 

14. The last line (355) may be missing a citation, as it says "(cite)" at the end. 

Author Response

Review 1

Comments and Suggestions for Authors

Thank you for the opportunity to review this submission. This article, entitled "LGBTQ+ Affirming Care May Increase Awareness and Understanding of Undetectable=Untransmittable among Midlife and Older Gay and Bisexual Men in the US South," assesses the relationship between having an affirming PCP and U=U awareness, belief, and understanding among older queer men in four southern states. Overall, the article is very well written, fills a gap in the literature, and has direct clinical and programmatic implications. 

Thank you for this review and for your time.

The following minor revisions would further enhance the quality and clarity of the article ahead of publication and dissemination: 

Overall: 

  1. Recommend using consistent language throughout, specifically around the age range of the population and terminology describing HIV status. The title and beginning of the article refer to "midlife and older" men, but then most of the tables and results only speak to "older" men. I would also recommend consistency around "living with HIV" - see line 33 for example of "people with HIV" or line 63 for example of "they have HIV" language. 

We have revised all instances of “older” that pertain to our results and study to “midlife and older” as this best represents the life stages of participants. Where we are citing other work, we have confirmed the usage in the original research. We have also revised all instances referring to people living with HIV as such for consistency as requested. Thank you.

Introduction: 

  1. Line 57 introduces "studies" demonstrating that HIV-negative men in the South are significantly less likely to have heard of U=U, however only one study is cited. Additional references are needed to support that multiple studies have demonstrated this finding, otherwise suggest revising the language here. 

Thank you. This statement referred to the following sentence as well, which was unclear. We have joined these two sentences to read:

“Additionally, among studies with adequate sample sizes to test geographic variation within the US, HIV negative men living in Southern states in the US were significantly less likely to have heard of U=U, [7] and men living with HIV in Southern states were less likely to rate the U=U concept as accurate compared to men living with HIV in the Northeast and Western states [11].”

Methods: 

  1. Line 132 - might suggest another term besides "use" of LGBTQ affirming care. The current language might imply affirming care is readily available and uptake is the issue. Possibly "receipt of" or "provision of" LGBTQ affirming care instead. 

Thank you. We agree and have revised the language here to reflect “access” in line with this and other reviewer comments.

  1. Typo in Line 149 with duplicate "to."

Revised.

  1. Additional information would be beneficial under the Affirming Care Measures section. How was an affirming PCP quantified or asked in the assessment? Were possible reasons for not having an affirming provider listed for selection, or were the responses unprompted? More clarity and specificity, similar to that included in the U=U measures section, would strengthen the article since these measures are a primary focus of analysis. 

Thank you. We have added additional detail on this item. Specifically, we have added the following at about line 174:

“To assess access to an LGBTQ+ affirming provider, Respondents were asked “Do you have an LGBT-affirming health care provider?” with response options: “Yes, they are my primary health care provider; Yes, I see them in addition to another health care provider; No, I don’t need or want an LGBT-affirming health care provider; No, I cannot find an LGBT-affirming health care provider in my area; I don't know; and No answer.” Respondents who reported “Yes” were coded as having access to an LGBTQ+ affirming health care provider. No answer was coded as missing. All others were coded as no.”

In this version of the item, no definition or example was given to define LGBTQ+ affirming care for the respondent. We have added this to the limitations.

Please note, we also thought it important to assess what things respondents were thinking about when they identified their provider as LGTBQ+ affirming. In follow-up waves of this survey currently in the field, we have asked additional items to clarify and identify behaviors or other symbolic actions or policies that respondents link to “affirming” care. This is now the topic of a dissertation and will be a fruitful area of research!  With about 70% of the wave complete at 80% retention, the key things that most recognize as important are using of inclusive language and perceived competency in LGBTQ health issues (not asking me to educate them on LGBTQ+ issues, not referring me out about my health problems, answering all my questions). Only about 20% reported that their provider, practice, or hospital always or often displayed LGBTQ+ friendly materials like pins or pamphlets. When this wave is completed in late 2022, we will be able to assess provider and patient determinants of affirming care more fully.

Results: 

  1. For all tables, clarity would be improved by left aligning the variable names and indenting subcategories as appropriate. For example, indenting the four state names under "State of Residence" or the educational attainment level under "Education." Might also recommend, at the discretion of the authors, that the overall and stratified N's be added to the headers (similar to Table 1) instead of the last row for Tables 2-7. 

We agree on the justification for the left-most column and submitted all Tables this way, but centering the categories seems to be journal convention. We have moved Ns to the column heading as requested. Thank you.

  1. Reduce the number of decimal places reported in the Tables, and be consistent between the Tables and in-text statistics in terms of reported decimal places.

We have reduced percentages to one decimal place in all instances. Odds ratios and confidence intervals are now all presented with 2 decimal places.

  1. In Table 1, prior HIV testing behaviors are reported for HIV-negative men only (n=441), but then the overall frequency is different (n=556). It's unclear where the additional 115 responses in the Overall frequency are coming from. 

The 441 reflects the number of HIV negative men who have ever tested for HIV out of 538 HIV negative men in the sample (this is what is listed at the column header). 441/538=82.0%

What we have interpreted this comment to pertain to is the change in sample size from Table 1 to the regression tables (2, 4-7). We have updated tables 1 and 3 to reflect the restricted analytic sample rather than the full sample of participants who identify as gay and bisexual men. The Ns here are 502 for HIV negative men and 131 for men living with HIV. Those who are “lost” at analysis are missing on independent variables in the model. We hope this addresses this comment.

  1. Inclusion of chi-square statistics in Table 3 would be beneficial, as only one chi-square test is reported in-text. Additionally, the percentage in the text (17.5%) is slightly different than the percentage in the table (17.7%). 

Thank you, we have added chi-square and independent samples t-tests to Tables 1 and 3 to show significant differences in sample characteristics by HIV status and by receipt of affirming care.

  1. If space/word count allows, consider speaking to other statistically significant relationships, as noted in the Tables, in the text. 

Thank you. We have added the following at the end of paragraph 1 of results (about line 195). “Men living with HIV were significantly more likely to have an LGBTQ affirming care provider, to be Black or other\multiracial compared to white, and to have less education than men who were HIV negative.”

Discussion: 

  1. Line 298 refers to findings from the broader literature, but no citations are included.

Thank you. We have added citations here.

  1. It would be beneficial to speak to the representativeness of the sample, either in the Results or the Limitations. For example, the sample included in the analysis is largely White, educated, and in the lower end of the eligible age range. Are data about recruitment strategies available to report? For example, were most men recruited from online platforms or in-person venues? Is there the potential for bias based on these data? 

We have added the following:

“The VUSNAPS panel generally reflects the demographic characteristics of the LGBTQ population in the 50 to 76 age range for sample states and the US South as measured by the US Census Household Pulse Survey (HPS), Phase 3.2, weeks 34-39 (see Appendix Table 1). Compared to weighted HPS estimates of demographic characteristics of LGBTQ people in the US South Census region, VUSNAPS participants are more educated, less likely to identify as bisexual, and less likely to identify as Latino/Hispanic.”

  1. The last paragraph (Lines 350-355), especially the last sentence, seem somewhat removed and generalized from the direct findings of the paper. Given the significant findings found from a region disproportionately burdened by HIV and with numerous disparities in terms of equitable and quality care, the currently worded closing paragraph seems like a missed opportunity to really highlight the implications, clinical or policy recommendations, and/or next steps from your research. 

We have moved the original final paragraph above the limitations paragraph (about line 366) and added the following:

“The Southern region also has the greatest burden of HIV-related deaths in the US. Compared with those unaware of U=U, people living with HIV who had U=U-related discussions with a health care provider had better odds of adherence to HIV treatments and viral suppression. [4] Expansion of LGBTQ affirming care could help reduce HIV-related mortality in the South by improving U=U awareness and uptake of HIV testing among HIV negative men.”

  1. The last line (355) may be missing a citation, as it says "(cite)" at the end. 

Thank you. We have resolved this omission.

Reviewer 2 Report

The paper is presenting widely the importance of awareness of U=U, of LGBTQ health care affirming providers  in HIV positive and HIV negative pts, It indicates the need of more medical education in terms of the topics mentioned above in certain groups of patients. It also indicates the need to obtain information about knowledge on U=U in younger or older HIV positive and HIV negative pts, including other data that can help to prepare education for certain groups of pts and  also health care providers. 

In the opinion of the reviewer the discussion should be developed containing more comparisons with other data.

Author Response

Review 2

[T]he discussion should be developed containing more comparisons with other data.

Thank you. We have added the following to the discussion:

“Awareness of U=U in this study is substantially lower than in other international and US surveys of people living with HIV and men who have sex with men that find high (70-90%) awareness of U=U [4,5,7]. Higher awareness in other studies may be due to sampling strategies that primarily engage individuals attached to organizations for PLWH or lack of disaggregation of HIV negative from men living with HIV in some samples. HIV negative and unknown status men who have sex with men are significantly less likely to be aware of U=U. [5]”

Reviewer 3 Report

This study investigated whether the LGBTQ+ affirming care increase awareness and understanding of undetectable=untransmittable among midlife and older gay and bisexual men in a large sample from the US south. The topic is important for the health care in the LGBTQ+ people. However, there are several methodology considerations to be clarified.

It is important to define “LGBTQ+ affirming care” earlier in the introduction.

In 2.2. Affirming care measures, the authors stated that “The VUSNAPS instrument includes several measures to assess healthcare access and utilization, including whether the respondent has a primary provider, whether the respondent’s primary care provider is LGBTQ affirming, use of a secondary healthcare provider that is LGBTQ+ affirming, and reasons for not having an LGBTQ affirming provider if no affirming provider is identified.” It is important to provide information how this information being collected. How is the instrument rated on these? Are they yes/no questions or open question? How these questions categorize the participants into LGBTQ+ affirming care or not? How to calculate on the items? Is there a cut-off point? Who rated on this instrument?

The statistical analysis methods need to be described specifically in the 2.4. Analytic strategy. The interaction between HIV positive/negative and LGBTQ affirming care or not needs to be examined in order to support the conclusion.

Regarding the findings that residence of state showed significance in believing or understanding U=U, the authors may discuss about the findings based on the policy or cultural backgrounds.

As for the implication, the authors stated that “Improving access to and provision of LGBTQ affirming care among sexual minority men may also reduce HIV stigma within the LGBTQ community.” Could the authors address more on how to improve the access to LGBTQ affirming care, such as health education, NGO, etc.?

Please consider the language use that might involve discrimination. For example,  “other person of color” in Table 1, “older gay”.

Please provide the information on research ethics committee for this study. 

Minor points.

Table Note. “+ p<0.1- *p<0.05- **p<0.01- ***p<.001”

Please replace ‘-‘ by ‘,’. Also, please be consistent in put 0 or not before the decimal point.

In the end of conclusion, there is a word of ‘(cite)’. What does it mean? Is there a missing reference?

Author Response

Review 3

This study investigated whether the LGBTQ+ affirming care increase awareness and understanding of undetectable=untransmittable among midlife and older gay and bisexual men in a large sample from the US south. The topic is important for the health care in the LGBTQ+ people. However, there are several methodology considerations to be clarified.

Thank you for this review and for your time.

It is important to define “LGBTQ+ affirming care” earlier in the introduction.

Thank you. We have provided the following definition: “LGBTQ affirming care—care that is respectful and meets the specific health needs of LGBTQ people—"

In 2.2. Affirming care measures, the authors stated that “The VUSNAPS instrument includes several measures to assess healthcare access and utilization, including whether the respondent has a primary provider, whether the respondent’s primary care provider is LGBTQ affirming, use of a secondary healthcare provider that is LGBTQ+ affirming, and reasons for not having an LGBTQ affirming provider if no affirming provider is identified.” It is important to provide information how this information being collected. How is the instrument rated on these? Are they yes/no questions or open question? How these questions categorize the participants into LGBTQ+ affirming care or not? How to calculate on the items? Is there a cut-off point? Who rated on this instrument?

Thank you. We have expanded the description of this measure given this and other reviewer feedback to be the following:

“2.2. Affirming Care Measures

The VUSNAPS instrument includes several measures to assess healthcare access and utilization, including whether the respondent has a usual source of care, whether the respondent has a primary or secondary health care provider who they perceive as LGBTQ affirming, and reasons for not having an LGBTQ affirming provider if no affirming provider is identified.

To assess access to an LGBTQ affirming provider, Respondents were asked “Do you have an LGBT affirming health care provider?” with response options: “Yes, they are my primary health care provider; Yes, I see them in addition to another health care provider; No, I don’t need or want an LGBT affirming health care provider; No, I cannot find an LGBT affirming health care provider in my area; I don't know; and No answer.” Respondents who reported “Yes” were coded as having access to an LGBTQ affirming health care provider. No answer was coded as missing. All others were coded as no.

The statistical analysis methods need to be described specifically in the 2.4. Analytic strategy. The interaction between HIV positive/negative and LGBTQ affirming care or not needs to be examined in order to support the conclusion.

Thank you. We agree that HIV status is an important factor for several of these outcomes. In addition, men living with HIV were significantly more likely to have what they considered to be an LGBTQ affirming provider. To ensure that HIV negative men with an affirming care provider were appropriately compared to HIV negative men without an affirming care provider (and, similarly, that men living with HIV with an affirming care provider were compared with men living with HIV without an affirming care provider), all analyses are already fully interacted by HIV status.

Regarding the findings that residence of state showed significance in believing or understanding U=U, the authors may discuss about the findings based on the policy or cultural backgrounds.

As for the implication, the authors stated that “Improving access to and provision of LGBTQ affirming care among sexual minority men may also reduce HIV stigma within the LGBTQ community.” Could the authors address more on how to improve the access to LGBTQ affirming care, such as health education, NGO, etc.?

Thank you. We have included the following:

“Additional clinical training and medical education courses on how to provide LGBTQ affirming care would likely decrease gaps in U=U awareness and understanding and may increase HIV testing among midlife and older HIV negative men in the US South. Other institution and practice level changes, such as the adoption of explicit nondiscrimination policies for patients and employees, and the use of inclusive language and images in health care settings may improve LGBTQ patient outcomes.”

Please consider the language use that might involve discrimination. For example,  “other person of color” in Table 1, “older gay”.

Thank you. We have updated all usage to “other\multiracial” in tables and text. We have discussed the use of older with our community and, although elder is also appropriate, elder had more objections. We have opted to stay with “midlife and older gay and bisexual men” as a descriptor of our sample. If additional suggestions are provided, we can review them with our community.

Please provide the information on research ethics committee for this study. 

Thank you. We have added: “This study was approved by the Vanderbilt University Institutional Review Board.”

Minor points.

Table Note. “+ p<0.1- *p<0.05- **p<0.01- ***p<.001” Please replace ‘-‘ by ‘,’. Also, please be consistent in put 0 or not before the decimal point.

We have corrected this in all instances.

In the end of conclusion, there is a word of ‘(cite)’. What does it mean? Is there a missing reference?

Thank you. We have resolved this omission.